# Development of culturally-appropriate text message booster content to follow a brief intervention focused on reducing alcohol related harms for injury patients in Moshi, Tanzania

**Catherine A. Staton**[1,2]*, **Deepti Agnihotri**[1], **Ashley J. Phillips**[1,2], **Kennedy Ngowi**[3], **Lily Huo**[1], **Judith Boshe**[3,4], **Francis Sakita**[4,5], **Anna Tupetz**[1,2], **Brian Suffoletto**[6], **Blandina T. Mmbaga**[1,3,4,5], **Joao Ricardo Nickenig Vissoci**[1,2]

1 Duke Global Health Institute, Global Emergency Medicine Innovation and Implementation Research Center, Duke University, Durham, North Carolina, United States of America, 2 Department of Emergency Medicine, Duke University School of Medicine, Durham, North Carolina, United States of America, 3 Kilimanjaro Clinical Research Institute, Moshi, Tanzania, 4 Kilimanjaro Christian Medical Centre, Moshi, Tanzania, 5 Kilimanjaro Clinical Medical University College, Moshi, Tanzania, 6 Department of Emergency Medicine, Stanford University, Palo Alto, California, United States of America

* catherine.staton@duke.edu

**Data Availability Statement:** The authors do not have permission to share the data widely according

## Abstract

Alcohol use is a risk factor for death and disability and is attributed to almost one-third of injury deaths globally. This highlights the need for interventions aimed at alcohol reduction, especially in areas with high rates of injury with concurrent alcohol use, such as Tanzania. The aim of this study is to create a culturally appropriate text messages as a booster to a brief negotiational intervention (BNI), to in the Emergency Department of the Kilimanjaro Christian Medical Centre, Moshi, Tanzania. Creation of text message boosters for an ED-based intervention expands the window of opportunity for alcohol use reduction in this high-risk population. The study followed a two-step approach to create the text message content in English and then translate and culturally adapt to Tanzanian Swahili. The culturalization process followed the World Health Organization's process of translation and adaptation of instruments. Translation, back translation, and qualitative focus groups were used for quality control to ensure text message content accuracy and cultural appropriateness. In total, nearly 50 text messages were initially developed in English, yet only 29 text messages were successfully translated and adapted; they were focused on the themes of Self-awareness, Goal setting and Motivation. We developed culturally appropriate text message boosters in Swahili for injury patients in Tanzania coupled with a BNI for alcohol use reduction. We found it important to evaluate content validation for interventions and measurement tools because the intended text message can often be lost in translation. The process of culturalization is critical in order to create interventions that are applicable and beneficial to the target population.

**Trial registration: Clinical Trials Registration Number**: NCT02828267, NCT04535011

to our regulatory agencies. As such, we can make the data available upon request to a third party Gwamaka William, gwamakawilliam14@gmail.com, at the agencies National Institute of Medical Research in Tanzania and the Kilimanjaro Christian Medical Center Ethics Committee.

**Funding:** This project was conducted with funding from the National Institute of Health Fogarty International Center K01- TW010000 (PI Staton) and the National Institute of Alcohol and Alcoholism PRACT: Pragmatic Randomized Adaptive Clinical Trial to Investigate Controlling Alcohol related harms in a Low-Income Setting; Emergency Department Brief Interventions in Tanzania R01AA027512 (PI Staton). The funders had no role in study design, data collection and analysis, decision to publish, or preparation of the manuscript.

**Competing interests:** The authors have declared that no competing interests exist.

## Introduction

Harmful alcohol use is linked to 3 million global deaths annually, a mortality rate that exceeds those of HIV/AIDS, malaria, and tuberculosis [1]. Similarly, injuries account for 5.8 million global annual deaths, making them a leading cause of global mortality [2]. Of the annual deaths caused by harmful alcohol use, 28.7% are caused by injuries [1]. The greatest burden of disease for both harmful alcohol use and injury rates is held by low- and middle- income countries (LMICs), specifically those in Africa [1–4]. Daily alcohol consumption is 20% higher in Africa than the global average [1]. Within sub-Saharan Africa, Tanzania experiences particularly high rates of harmful alcohol use and related injuries. The Tanzanian population consumes 9.4 liters per capita, compared to a regional Sub-Saharan African average of 6.4 liters per capita [1]. Further, the Kilimanjaro region also exhibits a higher prevalence of alcohol consumption compared to the rest of Northern Tanzania, potentially due to different reporting behaviors, socioeconomic status, and cultural practices [5,6]. At Kilimanjaro Christian Medical Centre (KCMC), a large referral hospital in the Kilimanjaro region, an estimated 30.0% of injury patients presenting to the ED test positive for alcohol [7]. Furthermore, Tanzania hosts a high prevalence of alcohol use disorders (AUDs) at around 6.8% [1]. This rate increases to 10.5% in the Kilimanjaro and Mwanza regions [5]. There is a need to reduce harmful alcohol use and injury rates in Tanzania.

Mobile health, or mhealth has emerged as a promising modality for healthcare delivery, especially in low-resource settings. In Tanzania, the popularity of mobile subscriptions and the accessibility of text messages over existing networks are increasing, with recent subscription rates in Tanzania averaging about 70 mobile-cellular telephone subscriptions per 100 inhabitants [8]. Systematic reviews suggest text message interventions can improve behavior management for health issues including diabetes, weight loss, physical activity, smoking cessation, antiretroviral medication adherence, and chronic diseases in LMICs [9–13]. Text message-based health interventions in the KCMC community have been successful in delivering text messages relating to vaccinations, medication adherence, and counseling [14–18].

As for alcohol use, text message have been shown to reduce alcohol consumption in a variety of population groups and settings, especially when combined with effective brief interventions, such as the brief negotiational interviews at the emergency department [19–23], yet those studies have been mostly been performed in high-income countries. Studies have also considered the use of personalized text message boosters to reiterate BNI text messages to increase their success with marginal additional costs [24]. While promising, text message boosters have not yet been translated and adapted to be used at KCMC, Tanzania or Swahili.

At KCMC, we have developed the BNI, "Punguza Pombe Kwa Afya Yako" (PPKAY)/ "Reduce Alcohol for Your Health," as a one-time, 15-minute, nurse-led motivational interview to reduce hazardous/risky drinking behaviors. [25] However, while text message boosters for alcohol-related BNI have been developed, validated and implemented in other countries, there is a gap in the literature about the content adaptations needed to apply similar text message boosters to this population. The development of an evidence-based text message booster may thus help improve the effectiveness of the intervention at reducing high-risk alcohol-related behaviors. The aim of this study is to describe the content design, culturalization, and validation process of a set of text message boosters used to supplement a BNI for alcohol use among acute injury patients who present at the KCMC ED.

## Methods

### Ethical statement

Kilimanjaro Christian Medical University College Ethics Committee (Research Proposal #497), Duke University Health System Institutional Review Board (Pro00062061), and the Tanzanian National Institute of Medical Research (Ref. NIMR/HQ/R.8a/Vol. IX/2121) all approved this protocol. Participants provided written informed consent. Our pilot study registration number is NCT02828267, and our ongoing clinical trial is registered with Clinical-Trials.gov as NCT04535011.

### Inclusivity in global research

Additional information regarding the ethical, cultural, and scientific considerations specific to inclusivity in global research is included in the (S3 Checklist).

### Setting

This study took place in KCMC in Moshi, Tanzania. Moshi has over 180,000 residents and has been the site of studies focused on implementing and testing the utility of M-health based interventions [17,26]. Text messaging has proven to be a feasible model of health information/education delivery and data collection in Moshi with an over 50% response rate [17].

### Study design

This study was a two-phase, mixed methods study to develop, translate, and validate the content of text messages for a BNI aimed at reducing harmful alcohol use between September 22, 2020 and October 5, 2020. The content for the text messages was developed using *Intervention Mapping*, a procedural framework that outlines steps for designing and implementing an intervention based on a theoretical foundation [27,28]. This framework was chosen to allow for reproducibility and to enhance rigor of our process. The steps for intervention mapping are 1) conducting a needs assessment, 2) identifying outcomes, performance objectives, and change objectives, 3) selecting theory-based intervention methods, 4) organizing methods into an intervention program, 5) creating an implementation plan, 6) creating an evaluation plan [27]. Our previous research established the need for a low-cost intervention for reducing harmful alcohol use including steps 1 and 2 as noted above [6,7]. The scope of this study encompasses steps 3 and 4 of the intervention mapping framework while steps 5 and 6 are further described in our pragmatic randomized adaptive clinical trial (Fig 1) [29]. This cultural adaptation process followed the World Health Organization's (WHO) Process of Translation and Adaptation of Instruments [30].

**Phase 1: Text message content development.** Given the promising efficacy of text message boosters to reinforce alcohol-related BNI text messages found in prior studies, a literature review was conducted to identify validated text messages that could be adapted for the KCMC setting. We conducted a snowball literature search to identify text messages that had been previously validated for use in alcohol reduction interventions. Because the aim of the overarching BNI was to promote safer drinking behaviors, we included validated text messages in our pool (updated December 2018) that specifically focused on themes of **self awareness, goal setting, motivation**. These three themes were specifically chosen because they encompassed necessary components of successful, sustainable behavior change [31]. The literature-based text messages on these themes were included in our preliminary English text messages pool, with edits made by our Tanzanian research team (KN, FS and BM) to any content that needed cultural appropriateness changes.

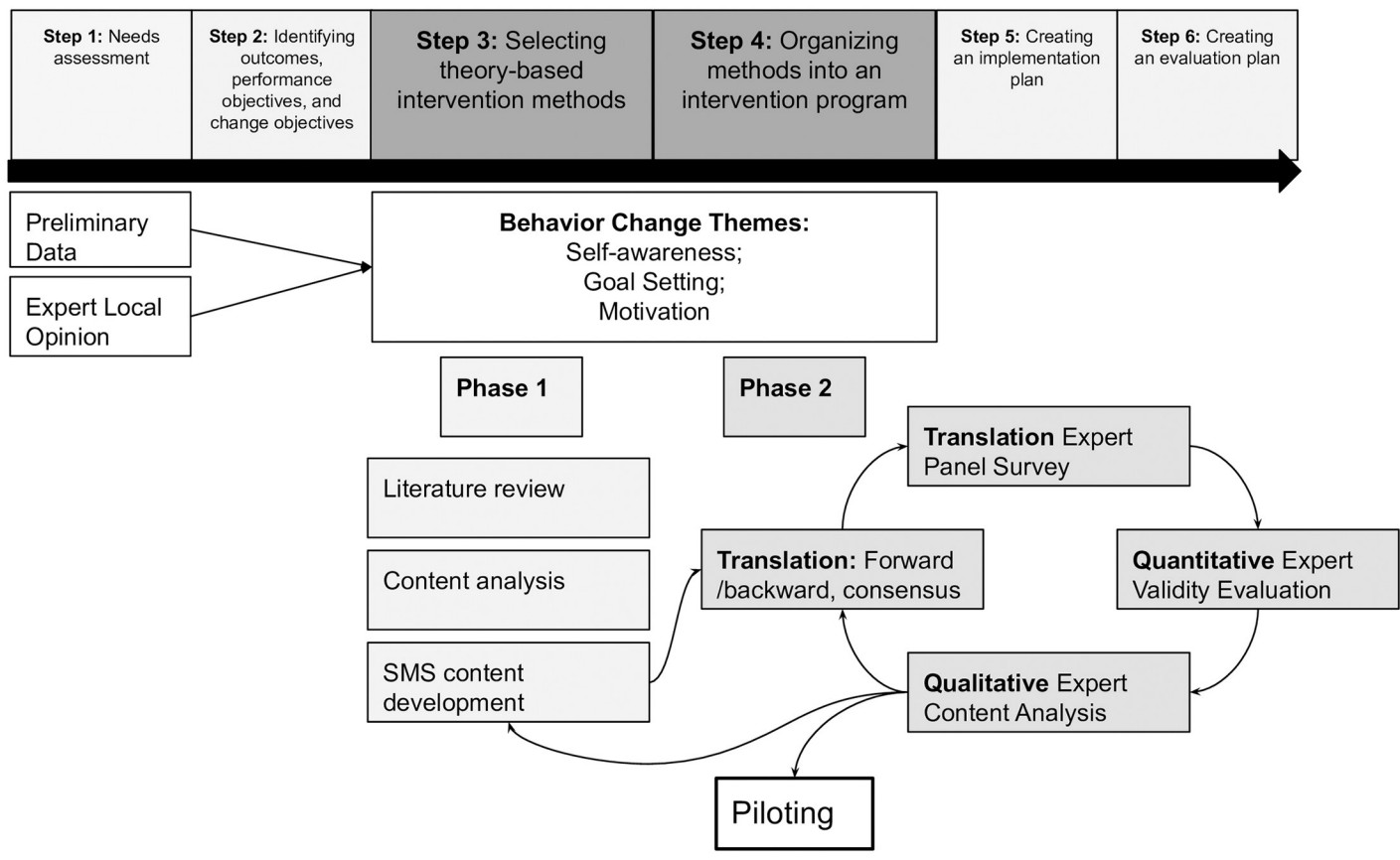

**Fig 1. Flow diagram of development and validation process.**

**Phase 2: Text message content validation.** In the second phase, our English text messages were translated to Swahili and evidence of validity related to the content of the text messages was obtained in a three-step process including: (a) translation, (b) content evaluation and (c) piloting. A combination of quantitative and qualitative methods were used to ensure that text message boosters were coherent to the theme of safer alcohol use by both adhering to thematic concepts and yet being culturally and linguistically appropriate.

*(a) Translation Processes*: One bilingual researcher translated the text messages from English to Swahili, with a second bilingual researcher back-translating the text messages to English to ensure the text messages conveyed the intended meaning. The researchers, individually then jointly, compared the original text messages with the back translated versions to identify discordances. If consensus could not be reached on how to adjust the translation, another pair of researchers were invited to translate and back-translate the text messages. At the end of the translation process, we had our first expert review, where two experts (FS and BM) reviewed the first version of the Swahili text messages and offered comments and recommendations.

*(b) Content evaluation processes*: Once a consensus was reached on the translations, we evaluated the content of the translated text messages with nine expert judges. The judges were physicians, professors and researchers fluent in English and Swahili with expertise in mental health, injury prevention, and psychology. The surveys contained questions about the clarity, domain fidelity and cultural appropriateness of the Swahili text messages. We asked expert

judges to rate each text message in terms of clarity, appropriateness to the Tanzanian culture, and fidelity to the alcohol reduction daily activities using a five-point Likert scale. In addition to the quantitative questions, the survey also contained an open-ended qualitative portion where experts could suggest edits to each of the items. We conducted two rounds of surveys, one with five experts and the second with four. In the first round, judges were asked to rank all the text messages. The research team reconvened to review the text messages with any rating lower than 4, evaluating any qualitative comments and making edits. A second version of the edited text messages only were submitted for the judges for rating.

After the surveys were conducted, our team revised all the Swahili text messages in a focus group with 5 of the bilingual experts judges (physicians, professors, nurses and researchers fluent in English and Swahili with expertise in mental health and psychology). Swahili text messages were read in Swahili and the group back-translated them and discussed the meaning and potential impact of the text messages. Then, participants suggested changes to Swahili content, (and back translated English content), in order to improve the text message in Swahili. This process was continued until consensus was reached in the discussion group, or if it was not, researchers suggested removing those from the pool of options. The group again determined if a text message would be removed because it was not relevant linguistically or culturally.

*(c) Piloting processes*: Finally, we conducted a piloting phase where we used our text message content on our planned Kilimanjaro Clinical Research Institute (KCRI) Short Message Service (SMS) System. First, we did this with our 10-person Tanzanian research team to 1) make sure texts were received and 2) ensure comprehension and ease of reading the text messages on the phone system. Next, we had potential patients receive these texts on their phones and with the help of our research team, and solicited feedback on the clarity, cultural appropriateness and potential impact of one of 10 text messages. We enrolled 15 patients at KCMC that had consented to receive the text message and evaluate their content clarity, cultural appropriateness, and impact. Each patient received the text message after consent, in front of the research assistant. After receiving the text messages, the patients would participate in a cognitive interview about their understanding of the text message content, and reply to a text message-based survey to suggest edits to the research team.

**Data collection and data analysis.**   Expert judge content validation was surveyed in English using REDCap, and responses were exported into Excel to be analyzed by a bilingual Tanzanian research assistant. Since our goal was to get a rating of content to bring to our qualitative focus group panel, our analysis of the survey was focused on identifying the Swahili text messages with cultural adaptation problems and suggested changes that were analyzed in the qualitative evaluation. Content analysis was conducted on the qualitative comments using inductive coding to create a list of recurring concepts or themes [32]. Analyses were conducted using Microsoft Excel. Each iteration of the text message (original draft and subsequent iterations) was saved on a spreadsheet on Microsoft Excel.

## Results

### Phase 1: Text message content development

Our literature review identified 10 studies with a primary focus on the development and evaluation of tailored text messages for alcohol use reducing interventions [10,12,19,33–47]. From these manuscripts, we accumulated about 30 initial text messages which could be adapted in order to match our three **self awareness, goal setting, and motivation themes** (See S1 Appendix). Of these three, we started to find sub-themes of control, competence and empowerment (self-awareness), intention and goal setting (goal setting), and motivation and commitment (motivation). We also found a significant number of fact text messages which were not

**Table 1. Sample text message content after the translation of standard and personalized components.**

| Standard Text Message | Standard Text Message, Swahili | Personalized Text Message | Personalized Text Message, Swahili |
|---|---|---|---|
| "Reducing your alcohol intake to [<5 drinks per occasion for men or <4 drinks per occasion for women] drinks per day reduces your risk of alcohol-related consequence" | "Kupunguza nywaji wako wa pombe mpaka (<vinywaji 5 kwa wanaume kwa kikao kimoja au <vinywaji 4 kwa wanawake kwa kikao kimoja) kwa siku inapunguza hatari yako ya madhara ya pombe" | "Remember try to keep your goal of less than [XX] drinks per occasion to achieve your goal of . . .[being a better husband]." | "Kumbuka jaribu kuweka lengo lako la vinywaji chini ya [XX] kwa kikao kimoja ili kufikia lengo lako la . . . [kuwa mume bora]." |
| "Having drank any amount of alcohol prior to driving a car or motorcycle has been shown to increase your risk of a road traffic injury. Stay safe, stay sober." | "Kunywa kiasi chochote cha pombe kabla ya kuendesha gari au pikipiki imeonesha kuongeza hatari yako ya kuumia barabara. Kuwa salama, usilewe. " | "Any amount of alcohol prior to driving a car or motorcycle has been shown to increase your risk of injury. Don't drink before you drive to achieve your goal of. . . .. [being there for my kids.] | "Kiasi chochote cha pombe kabla ya kuendesha gari au pikipiki imeonyeshwa kuongeza hatari yako ya kuumia. Usinywe kabla ya kuendesha gari ili kufikia lengo lako la . . . .. [kwa ajili ya kuwatunza watoto wangu.] |
| "Changing your routine or habits can help you reduce your alcohol intake." | "Kubadilisha utaratibu wako au tabia zako kunaweza kukusaidia kupunguza unywaji wako wa pombe. | "To achieve your goal of [.. . ..] change your drinking routine or habits. Even if it is just drinking one less drink per day" | "Ili kufikia lengo lako la [.. . ..] badilisha taratibu zako za unywaji au tabia zako za unywaji. Hata kama ni kunywa kinywaji kimoja au chini ya kinywaji kimoja kwa siku " |

pertinent in the Tanzanian setting and thus were excluded (e.g.,. "FACT: 2 drinks = 1 cheese-burger in calories. A lot of unnecessary calories!") As such, many of the fact-based text messages were removed, leaving 30 text messages to proceed.

## Phase 2: Text message content validation

*Text Message Translation and Back Translation*. back-translated. A sample of our back-translated text messages first version were created and shown in Table 1.

## Expert quantitative and qualitative validation

Survey comments submitted by the expert judge panel revealed that text messages needing revision mainly fell short in terms of **technical terms, cultural alignment, or clarity** (Table 2). In the first two iterations of evaluation of evidence of content validation, experts suggested

**Table 2. Common themes present causing difficulty with translation from English to Swahili.**

| Codes | Examples |
|---|---|
| Technical terms | • There is not a consensus from experts as to whether Tanzanians will understand the term "stress," but there is not a better word to describe it. |
| Cultural alignment | • The act of writing something down in order to remember it or making lists to remember things is not commonly practiced. All text messages that tell patients to write things down are not culturally appropriate. Instead, it is recommended to direct people to remember things rather than write them down<br>• Jogging is not something commonly practiced by people for joy or to reduce stress. Rather, an alternative statement should be used to encourage others to find something that makes them happy |
| Clarity | • The text messages need to be very specific. "Drinking" does not imply "drinking alcohol." The word for alcohol needs to be added each time drinking is mentioned<br>• Some of the text messages do not specifically relate to drinking (examples below). It is possible that some people might be confused as to why they are receiving a text message telling them to exercise or eat vegetables. It might be more challenging for them to infer that those activities are substitutes for drinking if it is not explicitly stated.<br>• Take control! Build up your strength by eating vegetables, drinking water, and sleeping. |

**Table 3. Example of revision process.**

| Initial translation | Edited version | Swahili version |
|---|---|---|
| Drinking too much can spoil a good night out and make you regret things. Stay safe, stay sober! | Drinking too much alcohol can spoil a good night out and make you regret things. Stay sober, stay safe. | Kunywa pombe sana kunaweza kuharibu mtoko mzuri wa usiku na kukufanya ujute. Usilewe, kaa salama. |
| Identify triggers that make you want to drink and try to find alternatives. Write them down. | Identify the triggers that make you want to drink alcohol and try to avoid them. Make a list so you can remember them. | Tambua vishawishi/vichocheo ambavyo vinakufanya utake kunywa pombe na jaribu kuviepuka. Viorodheshe ili kuvikumbuka. |
| Exercise! Endorphins gained will improve your mood and you will look and feel better too! | Do exercise! Exercise will lead to happiness, improve your health, help you look good, and help you feel better. | Fanya mazoezi! Mazoezi yatapelekea kuwa na furaha, yataboresha afya yako, yatakusaidia kuwa na mwonekano mzuri, na kukusaidia kujisikia vizuri. |

terms and phrasing that would strengthen the translation and clarify the meaning, such that it accurately corresponded to the English version. In terms of translation quality, there was a consistent disagreement between expert judges on the best way to translate the mental health construct of stress. While there is not a specific Swahili word for stress, the medical term in Swahili used 'mafadhaiko' is not easily understood by lay people. The suggestion was to keep stress in its English format that is more widely understood. As for cultural alignment, one of the text messages focused on known coping skills of taking notes or making lists to handle stress. It was discussed that this particular coping skill was not commonly practiced in the setting. A similar issue was observed with the lack of cultural appropriateness to suggest people could cope with the anxiety that lead to alcohol consumption by doing exercise, such as jogging. Finally, clarity issues were observed with the translation of the word 'drinking', that in Swahili would not imply 'drinking alcohol', the word alcohol was needed to clarify the sentence. Similarly, text messages need to be clear about being related to a coping mechanism to avoid drinking habits, to avoid confusion since some of the text messages were not commonly known coping skills.

From all of the Swahili text messages, 10 needed significant revision and discussion with a team of experts. Those text messages were flagged and specifically discussed during the qualitative discussions (sample of text messages edited are depicted in Table 3). During our qualitative discussion, our experts discussed the translation quality, cultural alignment and clarity while ensuring adherence to our initial thematic goals. By the end of the discussion, we either adapted text message content in order to be more appropriate or determined to remove the content if it was not appropriate. Ultimately, a total of 29 text messages completed the process and were retained in the process (S1 Appendix).

## Piloting

Piloting our 29 text messages with our research team was completed with the team's personal phones. Each of our 10 research assistants reported receiving all intended text messages and that text messages were clear and the SMS technology worked. During our patient piloting phase, we enrolled 15 patients. During this process, we received qualitative feedback from our research team, who introduced the text message pilot to the pilot participants, that the text message content was appropriate and well received. The SMS system, which was supposed to be sending text message content followed by questions for rating of clarity and perceived impact per text message, however, was not working. Participants would receive the survey questions at irregular times and order, and were not able to send information back to the system appropriately. Even with the research team's assistance, the text message survey response

data was not able to be appropriately collected. Our research team was able to discuss each text message sent to participants and received feedback that there were challenges with clarity, cultural alignment, nor translation quality. This problem was informative of the data collection through SMS system and improvement needed, but that was not part of the study of the intervention.

## Discussion

This paper describes our process of a rigorous content adaptation of evidence-based text messages to support alcohol harm reduction brief intervention in a limited resource and low income setting. Adapting alcohol reduction content from high-income settings to different resources, culture and languages requires an in-depth, structured and iterative process. Our two phase *Translation* and *Validation* iterative process incorporated both quantitative and qualitative methods in order to ensure cultural and linguistic translation while maintaining adherence to the needed themes and domains needed for the intervention. Our mixed method process identified numerous areas for improvement in the translation of technical terms, cultural alignment and clarity of language which was implemented after group discussion with experts. Overall, this process highlights the need for 1) an in-depth and evidence-based culturalization process which incorporates cultural (Tanzanian) and topical experts, and 2) a mixed method of data collection to accurately identify areas for improvement and iteratively improve text message content.

### Culturalization requires a rigorous iterative process

As we initiated this project, we planned to adapt evidence based practices used in high-income settings using the World Health Organization's (WHO) Process of Translation and Adaptation of Instruments. [30] We expanded this WHO process to be an iterative process focusing on both the language and cultural translation of the text messages. Even with this rigorous framework, there are multiple expert, evidence-based or patient-centered decisions to be made during this process, which inform both subsequent processes and end products. For example, Sharpe et al found it most appropriate to offer text message content in multiple languages based on their context specific findings. In contrast, since our setting uses a national Swahili language and most of the population is comfortable in Swahili but not English, we chose to focus only on a Swahili language and cultural translation. This was a specific choice given our preliminary data demonstrating nearly 99% of our population was Swahili speaking; that said, the generalizability of our intervention outside of the Kilimanjaro region of Tanzania might be more limited.

As stated in the WHO process, aside from linguistic translation, cultural tailoring and validation is very important. The type of information in the content must be culturally and locally relevant, creating a sense that the information provided is relevant to the individuals receiving the text messages. For instance, Wright et al found feedback that generic text messages, including normative feedback irrelevant as participants felt they knew about norms [44]. In stark contrast, our setting, unlike Wright's, has very limited health literacy or alcohol health literacy and a steep integration of alcohol use in cultural behaviors normalizing use with little public health education about the dangers of alcohol. For instance, in our preliminary work we found women use alcohol for its caloric content during pregnancy and for children, unaware of potential consequences [6]. Similarly, drinking driving is an unfortunately common challenge in commercial driving in the region with limited alcohol harm prevention education, alcohol regulation or police intervention or other protective factors [6]. Given this, in our setting, normative content would be beneficial but it is not yet relevant to the public. Unfortunately, given

the limited relevance of fact containing text message content to our setting, and limited literature for appropriate translation of these facts to pertinent data, all fact based text messages were excluded. On the other hand, we learned that a very specific reference to alcohol use is needed to link the text message's content to the drinking behavior.

All that being said, adherence to the tenets of the intervention, and in our case the BNI and themes chosen for text message content is key. Motivational interviewing concepts are uncommon in our Tanzanian healthcare setting but are a key component of our BNI and text messaging intervention; as such, we prioritized motivational text messages adapted and translated to this context. This coincides with literature supporting that participants prefer text messages which are motivating and positively framed and avoid warnings and statements not to drink [47].

## Mixed method approach allowed for a more in-depth explanatory process

As we had encountered 50 potential text messages, we sought to have both a validation as well as a way to prioritize which text messages were most appropriate/ preferred for cultural relativity. We included a mixed method approach in or to both 1) identify text message content which might be problematic for a more indepth review and 2) attempt to highlight which text messages might be more impactful to reduce alcohol use holistically, similar to Toyama's et al. method [48]. While our RedCap based quantitative expert data collection process was a success in highlighting text messages for further review, our patient facing SMS system based data collection process was not. We feel that this was a combination of the system delays, internet delays at the hospital where we enrolled potential patients and the overall limited technological comfort with this method of data collection. We do believe that our qualitative process did adequately provide information holistically on the translation and validity of our text messages but we were not able to rank text messages to remove ones deemed not as potentially impactful.

## Limitations

While we believe this process yielded culturally and linguistically valid text message content, there are some limitations which must be acknowledged. First, as we adapted evidence based messaging from high income settings, it is possible that we did not incorporate some potentially important messaging content had our methods been to create *de novo* text message content in Swahili and with locally derived themes. We chose our methods because, at the time of our project, we had no Tanzanian Swahili-speaking psychiatrist or alcohol practitioner in the Kilimanjaro region to lead this charge. We believe that this adaptation process provided an evidence-based framework to start from and further text message content can be created, added or changed based on Tanzanian expertise and participant feedback. Next, given an overall limitation in human resources at our site, during subsequent intervention development periods of this process, we had gained only one bilingual psychiatrist and alcohol use disorder practitioner who could provide feedback and be available to assist in serving as an expert. That being said, we have incorporated nearly a dozen Tanzanian counselors, nurses, physicians and researchers, as well as international experts including a psychologist and researchers to ensure appropriate cultural translation. We believe that our methods might actually have improved the population level linguistic relevance accounting for limited literacy and more common verbiage and phrases. Similarly, while our overall numbers for our quantitative processes are limited, we are more interested in areas of discordance using the quantitative processes to identify messaging which needs further quantitative evaluation; rendering the limited sample size less impactful. Finally, our piloting did identify challenges with our SMS system in our planned

data collection process as such, we will use different processes for future data collection. Similarly, we will reduce the need for participation and interaction in the SMS system during deployment of our intervention to reduce process complexity.

## Conclusion

Stemming from an intervention mapping and WHO Process of Translation and Adaptation of Instruments frameworks, we developed a rigorous mixed method process of translation and validation of text message boosters to support reducing harmful alcohol use. Adaptation of evidence-based content to a low resource setting requires both a timely and in depth process with local cultural and topical/alcohol harm reduction experts. We found that a mixed method iterative process with multiple rounds of expert evaluation followed by piloting found areas of improvement in translation quality, cultural relevance, clarity while allowing adherence to our chosen themes of self-awareness, goal setting and motivation. While a de novo creation of text message content might have different efficacy, adaptation of evidence-based messaging with linguistic and cultural validation is more feasible for limited resource settings.

## Supporting information

**S1 Checklist. GUIDED checklist.**
(DOCX)

**S2 Checklist. TIDieR checklist.**
(DOCX)

**S3 Checklist. Inclusivity in global research questionnaire.**
(DOCX)

**S1 Appendix. SMS versions and feedback summaries.**
(XLSX)

**S1 Protocol. Clinical trial protocol.**
(DOC)

## Acknowledgments

### Funding

This project was conducted with funding from the National Institute of Health Fogarty International Center K01- TW010000 (PI Staton) and the National Institute of Alcohol and Alcoholism PRACT: Pragmatic Randomized Adaptive Clinical Trial to Investigate Controlling Alcohol related harms in a Low-Income Setting; Emergency Department Brief Interventions in Tanzania R01AA027512 (PI Staton).

## Author Contributions

**Conceptualization:** Catherine A. Staton, Blandina T. Mmbaga, Joao Ricardo Nickenig Vissoci.

**Data curation:** Ashley J. Phillips, Kennedy Ngowi, Blandina T. Mmbaga.

**Formal analysis:** Deepti Agnihotri, Joao Ricardo Nickenig Vissoci.

**Funding acquisition:** Catherine A. Staton, Blandina T. Mmbaga.

**Methodology:** Deepti Agnihotri, Joao Ricardo Nickenig Vissoci.

**Project administration:** Ashley J. Phillips.

**Supervision:** Catherine A. Staton, Judith Boshe, Francis Sakita, Blandina T. Mmbaga.

**Visualization:** Anna Tupetz.

**Writing – original draft:** Deepti Agnihotri, Lily Huo, Anna Tupetz.

**Writing – review & editing:** Catherine A. Staton, Ashley J. Phillips, Kennedy Ngowi, Judith Boshe, Francis Sakita, Anna Tupetz, Brian Suffoletto, Blandina T. Mmbaga, Joao Ricardo Nickenig Vissoci.

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
