## [Decision Letter · Decision Letter 0]

5 Mar 2024

PGPH-D-23-02241

Development of culturally-appropriate text message booster content to follow a brief intervention focused on reducing alcohol-related harms for injury patients in Moshi, Tanzania

Dear Dr. Staton,

Thank you for submitting your manuscript to PLOS Global Public Health. After careful consideration, we feel that it has merit but does not fully meet PLOS Global Public Health’s publication criteria as it currently stands. Therefore, we invite you to submit a revised version of the manuscript that addresses the points raised during the review process.

We look forward to receiving your revised manuscript.

Kind regards,

Kathleen Bachynski, PhD, MPH

Academic Editor

Journal Requirements:

Additional Editor Comments (if provided):

Thank you very much for submitting your manuscript to PLOS Global Health. The manuscript addresses an important area of research in global health. All three reviewers recommend revisions to further strengthen the manuscript, including greater clarity on the methods and results, and addressing limitations such as the lack of an English–Swahili language expert and patient or community feedback on the messages. Therefore, I invite you to respond to the reviewers’ comments and revise your manuscript.

Reviewers' comments:

Reviewer's Responses to Questions

**Comments to the Author**

1. Does this manuscript meet PLOS Global Public Health’s publication criteria? Is the manuscript technically sound, and do the data support the conclusions? The manuscript must describe methodologically and ethically rigorous research with conclusions that are appropriately drawn based on the data presented.

Reviewer #1: Yes

Reviewer #2: Partly

Reviewer #3: Partly

2. Has the statistical analysis been performed appropriately and rigorously?

Reviewer #1: N/A

Reviewer #2: No

Reviewer #3: N/A

3. Have the authors made all data underlying the findings in their manuscript fully available (please refer to the Data Availability Statement at the start of the manuscript PDF file)?

Reviewer #1: Yes

Reviewer #2: No

Reviewer #3: No

4. Is the manuscript presented in an intelligible fashion and written in standard English?

Reviewer #1: Yes

Reviewer #2: Yes

Reviewer #3: Yes

5. Review Comments to the Author

Reviewer #1: The manuscript is well-prepared and does not need statistical analysis due to the nature of data used. It meets the PLOS Global Public Health’s publication criteria since it has included all important sections and is clearly written.

Reviewer #2: Thanks for giving me the opportunity to review this manuscript which is on adaptation of existing messages towards the Tanzanian setting. While interesting to read and certainly needs to be published, I am not able to get a complete picture of the methods and the related results. Reading it in the current format will not allow me or anyone else to reproduce what was done. I have quite a few questions (below) and hope that answers to the questions will help you to improve the manuscript.

Title: The title seems very long to me and I do not understand it well. I would remove ‘to follow a brief intervention’ or use a different wording

Abstract: The abstract in itself does not give me a clear view of the study. Reading it as a standalone abstract raises many questions. It is not clear to me what is meant by ‘culturally appropriate’.

Abstract introduction: I would expect more on the BNI. The sentence ‘Creation of text message boosters for an ED-based intervention expands the window of opportunity for alcohol use reduction in this high-risk population.’ does not add to the abstract as it is not clear how. Also, the abbreviation ED is not written out anywhere.

Abstract methods: I am not sure how the methods lead to cultural appropriateness. There is no explanation on who was involved in the methods and how analyses was done.

Abstract results: ‘yet only 29 text messages were successfully translated and adapted’. It is not clear why only 29. Were the others simply not done? Or were they not appropriate? If so, who decided on that?

Abstract conclusion: ‘We developed culturally appropriate text message boosters in Swahili for injury patients in Tanzania coupled with a BNI for alcohol use reduction.’ This abstract is not about developing the BNI. I would leave that out. The sentence ‘We found it important to evaluate content validation for interventions and measurement tools because the intended message can often be lost in translation.’ Seems overdone. In this abstract there is nothing about measurement tools.

Introduction

-Line 75: ‘Further, the Kilimanjaro region also exhibits a higher prevalence of alcohol consumption compared to the rest of Northern Tanzania, potentially due to different reporting behaviors, socioeconomic status, and cultural practices’. I would expect an explanation on how.

-Line 77: ED is used, but there is no explanation on what ED is

-mHealth: the term seems to be ‘old-fashioned’. I would change it to a more modern term such as digital health.

-Line 91: Text Message should be Text messages

-Page 4, paragraph 1: more information is needed on BNI

-Page 4, paragraph 2: more information is needed on what a text message booster entails

METHODS

This study is mentioned to be a mixed-methods study, but I do not see any quantification of results.

-Line 113: ‘Our pilot study registration number is NCT02828267, and our ongoing clinical trial is 114 registered with ClinicalTrials.gov as NCT04535011.’ The described study is not a pilot study, neither a trial. How does the described study fit with these two mentioned studies?

-Line 111: It is not NIMR who approves the proposal, it is the NATHREC based at NIMR. Also, for KCMC, it is CRERC who approves the proposal.

-Line 120: Reference 17 is according to the title of that publication not about education, but purely about data collection in a very specific group of people. Other studies in Moshi, among patients, have shown much higher percentages.

-Line 124: Are the text messages for the BNI or meant as a booster?

-In the abstract, it looks as if messages were found to be themed on self-awareness, goal setting and motivation. But in the methods the themes part of the selection criteria. Therefore, in the abstract, it should also be under methodology and not under results.

-Line 143: Why were members of the research team editing the message pool and not end-users (like those who drink alcohol and are ending in ED)? How are the research members representative? What are their functions/positions and why were they the ones working on that?

-Line 149: What is the definition of cultural appropriate changes?

-Page 7, 2nd paragraph: Why the judges were not lay end-users? I would expect that they are much more knowledgeable about understanding of messages than the experts.

-Line 168: ‘The surveys contained…’ Which surveys? Were surveys used to evaluate the content? If that is the case, it should be added to the sentence of line 166. Also, were the questions open or closed-ended?

-Line 188: What is the ‘planned Kilimanjaro Clinical Research Institute (KCRI) Short Message Service (SMS) System?’ How does it work? After reading the referred publication of Pima et al., I assume it is the same system? Then it could be described as such.

-Line 202: What is the definition of cultural adaptation problems?

-What were the ethical considerations? Was informed consent used for all steps?

RESULTS

-Line 212: It is not clear how subthemes were identified. Could this be elaborated on in the methods?

-Table 1: The table shows personalized messages, but it is not clear how these were created and why.

-I do not completely agree that cheeseburgers are not part of TZ culture. Cheeseburgers can be found more and more on different menus, including at local places.

-In the methods, it is described that three members of the research team worked on cultural appropriateness. How was this done in an objective way and what were the results obtained from this step?

-The paragraph from line 226 is very informative and interesting. Though reading, this implies that some kind of qualitative data was collected leading to these results though the first sentence mentions that these results came from a survey.

-Line 250: Why were only 29 completed in the process? What happened to the other messages?

-Line 260: The SMS system was not working… I wonder if it makes sense to mention this system at all in the manuscript. What does it still add to the manuscript while it wasn't used? If left in, a clear explanation is needed on why it didn’t work as despite of the sent SMSs of the BNI working, it will also raise questions about the feasibility of that.

DISCUSSION

-Line 295: Sentence starting with ‘For instance,….’ seems to be grammatically incorrect.

-The fact that the SMS system for data collection was not working seems surprising to me. Reading the publication from Pima et al ( that has been referred to several times) that seems to come from the same institute (KCRI?), there were no problems with such data collection. Again, I wonder to what extent mentioning this adds to the current manuscript.

GENERAL

The manuscript should align with the following criteria:

1. The study presents the results of primary scientific research. - YES

2. Results reported have not been published elsewhere. - YES

3. Experiments, statistics, and other analyses are performed to a high technical standard and are described in sufficient detail. -NOT YET. MORE DETAILS ARE NEEDED ON HOW ANALYSES OF DATA WAS DONE

4. Conclusions are presented in an appropriate fashion and are supported by the data. - YES

5. The article is presented in an intelligible fashion and is written in standard English. - YES

6. The research meets all applicable standards for the ethics of experimentation and research integrity. – NO. IT IS NOT CLEAR HOW INFORMED CONSENT PROCEDURES WERE DONE

7. The article adheres to appropriate reporting guidelines and community standards for data availability. - YES

Reviewer #3: This is a very well written article on the methods used for cultural adaptation and translation of a SMS text-messaging program to reduce problematic alcohol use in Tanzania. The process of cultural adaptation of the messages after the translation process is described in meticulous detail. However, there are three concerns I have with the current manuscript:

1. The process of testing the messages with prospective patients is under reported, and as there appeared to be major technological issues with this, this omission is a significant problem. Using the CONSORT-eHealth (https://pubmed.ncbi.nlm.nih.gov/22209829/) or mERA guidelines (https://www.bmj.com/content/352/bmj.i1174?etoc=) would improve the reporting of the test interventions used to test messages.

2. It is unclear from the writing if any feedback was collected from patient. On page 12, line 265, it states that the research team was able to discuss each text message, but the rest of the manuscript does not report any notes or analysis of these discussions. If the discussions occurred, some report of the methods used to gather information should be reported, even if informal. If this is a type, it should be clarified, adn the lack of patient feedback should be added to the discussion and limitations.

3. The lack of patient or community feedback on the messages is a potentially a fatal flaw; this adaptation of messages from high income countries to LMIC did include local clinicians, but the feedback from community members or patients is necessary prior to implementation; with out this, the potential benefit maybe substantially del

---

## [Decision Letter · Decision Letter 1]

25 Jun 2024

Development of culturally-appropriate text message booster content to follow a brief intervention focused on reducing alcohol-related harms for injury patients in Moshi, Tanzania

PGPH-D-23-02241R1

Dear Dr. Staton,

We are pleased to inform you that your manuscript 'Development of culturally-appropriate text message booster content to follow a brief intervention focused on reducing alcohol-related harms for injury patients in Moshi, Tanzania' has been provisionally accepted for publication in PLOS Global Public Health.

Best regards,

Kathleen Bachynski, PhD, MPH

Academic Editor

Reviewer Comments (if any, and for reference):

Reviewer's Responses to Questions

**Comments to the Author**

1. If the authors have adequately addressed your comments raised in a previous round of review and you feel that this manuscript is now acceptable for publication, you may indicate that here to bypass the “Comments to the Author” section, enter your conflict of interest statement in the “Confidential to Editor” section, and submit your "Accept" recommendation.

Reviewer #1: All comments have been addressed

Reviewer #2: All comments have been addressed

2. Does this manuscript meet PLOS Global Public Health’s publication criteria? Is the manuscript technically sound, and do the data support the conclusions? The manuscript must describe methodologically and ethically rigorous research with conclusions that are appropriately drawn based on the data presented.

Reviewer #1: Yes

Reviewer #2: Yes

3. Has the statistical analysis been performed appropriately and rigorously?

Reviewer #1: N/A

Reviewer #2: N/A

4. Have the authors made all data underlying the findings in their manuscript fully available (please refer to the Data Availability Statement at the start of the manuscript PDF file)?

Reviewer #1: Yes

Reviewer #2: No

5. Is the manuscript presented in an intelligible fashion and written in standard English?

Reviewer #1: Yes

Reviewer #2: Yes

6. Review Comments to the Author

Reviewer #1: The manuscrit has been improved, except that it needs more typographical errors correction and language polishing before publishing

Reviewer #2: All comments have been addressed, but I agree with reviewer 2 and 3 that it is a flaw that end-users were not involved in the design process. Knowing the Tanzanian community well, them not giving comments on SMS might also be a sign of either not understanding, or respect towards the research team or simply just not wanting to comment negatively. The latter is part and parcel of the culture. I would like to see this flaw described more clearly as I am not convinced about it now.

The second point relates to the data being made available. The Tanzanian procedures are to arrange a Data Transfer Agreement Form. This DTA should be completed by the provider (KCMC) and the receiver (the journal) and approved by the National Ethics Committee.

7. PLOS authors have the option to publish the peer review history of their article (what does this mean?). If published, this will include your full peer review and any attached files.

**Do you want your identity to be public for this peer review?** For information about this choice, including consent withdrawal, please see our Privacy Policy.

Reviewer #1: No

Reviewer #2: No
